# Promising Emerging Technologies for Teaching and Learning: Recent Developments and Future Challenges

Ahmad Almufarreh  and Muhammad Arshad *

Deanship of e-Learning and Information Technology, Jazan University, Alburj Campus,
Jazan 82812, Gizan, Saudi Arabia; almufarreh@jazanu.edu.sa
* Correspondence: msarshad@jazanu.edu.sa

**Abstract:** As time goes on and the number of people who use information and communication technology (ICT) grows, emerging technologies are receiving a lot of attention from academics, researchers, and users. Emerging technologies are changing more quickly than ever, and it is important to start building skills early in education by using the right examples to teach and learn about digital citizenship. New pedagogies support individual teaching and learning methodologies for knowledge acquisition and self-directed learning. Hundreds of digital education tools have been developed to increase student autonomy, enhance academic process management, promote collaboration, and improve communication between teachers and students. This article examines the current state of emerging technologies from a variety of educational viewpoints, highlights a few of them, and discusses both their established and potential educational benefits. Additionally, it offers in-depth debates on recent developments and future challenges from a wide range of perspectives. The analysis focuses on peer-reviewed research articles published in well-renowned publications databases over the last 8 years, drawing upon a bibliometric analysis using VOSviewer. The results of this study are also anticipated to serve as a substantial impetus for other universities and colleges worldwide to utilize innovatively and revolutionized technology for better academic performance.

**Keywords:** emerging technologies; teaching, and learning; higher education; augmented reality; virtual reality; artificial intelligence; internet of things; cloud computing

## 1. Introduction

Learners today have easier access to endless amounts of information and are subject to more distractions than ever before. How is Teaching and Learning (T&L) supposed to compete with social media, mobile applications, and the Internet? The goal is to take advantage of developing technological trends influencing teaching and learning and the T&L profession rather than compete. People can mix several learning strategies to produce something that takes advantage of the changing environment for teaching and learning. Everyday living involves using new software and technology, and being digitally literate is essential for participation in today's society and economy. Due to its potential in education, digital technology has sparked a greater interest in the topic. Theoretical frameworks for teaching and learning are challenged by learning in a world of new technology. A significant paradigm shift, enabled and driven by technological advancements, is the shift from instructor-based to learner-based education. Low-cost mobile technology has generated a great deal of curiosity and experimentation due to its affordability. The difficulty lies in utilizing modern technology effectively while ensuring that students are ready for successful lives in the twenty-first century [1].

The use of e-learning tools and technologies in third-world nations with low literacy rates is nothing short of a blessing. On the other hand, tools and technologies can meet the needs of all educational levels, regardless of place and time. A significant number of people may now learn using e-learning tools and technology, which also makes classes more participatory and save time and effort. E-learning tools and technology have advanced to help

a huge number of people study; in addition, they make classes interactive and save time and effort for learning [2]. The e-learning tools and technology are universally applicable in terms of efficiency, efficacy, and affordability. This resource was being utilized by the teacher and the student. As is well-known, traditional teaching involving a significant investment in infrastructure and money, both of which are frequently lacking. Additionally, the availability of quality teachers in comparison to the large population is a roadblock in education [3]. The well-being of students is significantly influenced by the quality of their instructors throughout their life, not just in terms of their academic performance but also in terms of other long-term social and occupational consequences.

All areas of work, the nature of occupations, tasks, and skills, as well as how people learn and live, are being impacted by digital transformation. In this article, the term "emerging technologies" refers to a broad range of products and services that make use of artificial intelligence (AI), virtual reality (VR) and augmented reality (AR), wearable technology, such as head-mounted displays and sensors, and the Internet of Things (IoT) made possible by the blazing-fast 5G mobile standard. With the help of these and other as-yet-undiscovered technologies, new kinds of digital ecosystems are developed based on the data generated by the numerous online interactions between people and their gadgets [4].

According to the literature, there have been several studies focusing on the utilization of emerging technologies in teaching and learning. The goal of this study [5] is to present a clear and methodical examination of cutting-edge research into the actual usage of contemporary technologies in foreign language classes, as it is supplied by the empirical research that is currently available globally. This study [6] highlights the connections and relationships between several broad difficulties related to the development of digital abilities as a necessary prerequisite for lifelong learning. Because technology is evolving at a quicker rate than ever, it is essential to start building skills early in education and include relevant examples for the development of digital citizenship abilities. The authors of [7] concentrate on adaptative learning technologies based on the perspectives of profound learning, where the achievement of objectives will be shown through created learning analytics, whose association may form consistent verifiable blockchains. This study [8] attempts to classify the various developing technology types, assess the degree of technology integration in the classroom and look at how technology affects creativity in the body of existing literature. In addition, this study looks at how the studies were planned to look into the usage of new technology to foster creativity and what key conclusions they came up with. This study [9] was inspired by the need to have a comprehensive understanding of research on creativity in settings that incorporated digital educational technology. The use of such technologies in education must take into account all learning/teaching concerns particular to a field as well as pedagogical considerations. Technology itself has no place in education. This study [10] seeks to promote higher engineering education, including policymakers, instructors, researchers, administrators, and students, to engage in critical thought regarding the datafication and digitalization of education, in which emerging technologies play such a significant role. These technologies are essential to addressing the urgent problems with the use of big data in the current educational system, instructional practices, ethical obligations, and sociocultural aspects of education. In this article [11], the study technique of a technologically improved approach to nurturing creativity was provided. The techno-self enhanced learning offers a learner-centered approach to creative learning by incorporating the new extended reality technologies. Then, to engage learners and further develop new methods of engagement, the cultural sector interpretation strategy was examined. Finally, the agile project development methodology was implemented. The literature on emerging technologies in the area of science education is reviewed in this article [12]. The authors analyze publications that have recently appeared in prominent scientific education forums to summarize the state of research today and identify particular sorts of technology that have lately "emerged" in K–12 science classrooms. There is a claim that new technologies do not genuinely develop in a sociocultural vacuum and that sociocultural aspects of technological innovation need to be discussed more in science

schools. To examine how multimedia technologies have proven to be a real strategy for bridging the gap in the provision of unrestricted access to quality education and improved learner performance, this paper [13] provides a systematic review of various multimedia tools in the teaching and learning processes. The study concluded that the majority of multimedia solutions used for teaching and learning target the pedagogical content of the subject of interest and the solution's user audience, while the technologies and components built into the development of the various multimedia tools have contributed to their success when used on various target groups and subjects. This study [14] examines the degree to which higher education institutions are currently prepared for the digital transformation of their operations. The study's goal is to look into the comparative measures that have been put in place and the difficulties that higher education institutions are having in dealing with the digital transformation of their operations. This study's particular significance relates to the extent to which administrative tasks, advanced communication between institutions, students, academic staff, and administrative staff, as well as other internal and external networking processes, are made possible by the use of digital technologies during the teaching process.

This study is being conducted to investigate cutting-edge technology for teaching and learning. This effort is motivated by the requirement for developing technologies since the teaching and learning environment will eventually need to support trillions of devices, software programs, and hardware components. According to this study, emerging technologies have the potential to significantly improve teaching and learning methods since they allow for more individualized learning and teaching experiences. Students can practice writing with emerging technologies in higher education, for instance, with the added benefit of growing an understanding of the audience they are writing for. The multi-phased search, evaluation, and analyses of research publications on emerging technologies for this study were directed by the following research questions:

1.  What are the promising emerging technologies for teaching and learning?
2.  What are their educational benefits, as reported in eligible research publications?
3.  What are the most recent advancements in emerging technologies for education, and what obstacles lie ahead?

With the pandemic phase, the new normal condition gives more educators a chance to experiment with cutting-edge technologies such as Telepsychiatry, Artificial Intelligence, Computer Vision, and Cloud Computing. Such research investigations are expected to be crucial in helping educators recognize the advantages of new technologies and actively incorporate them into learning environments.

## 2. Theoretical Concept

A good theory can be incredibly useful, but it can be very challenging to develop. To successfully predict fascinating experiences, it makes use of the appropriate notions, combining them in a minor way. According to Popper [15] a good theory can never be proved true but should be capable of being proved false. Good theories endure and are still useful because they enable people to comprehend education and behave morally. These ideas are still relevant today because new technologies and practices are frequently used to address the same issues and challenges that motivated researchers and educators to work with earlier technologies, which, while now well-established, were previously emergent. New educational technologies are fundamentally about change [16]. Both the technological environment and the learning environment are changing as a result of this transition. The theory of change states that until a change happens, there is a state of equilibrium in which all forces that could drive change are equal to any forces resisting change. For change to occur, the equilibrium's balance must be upset. This imbalance can be achieved by either enhancing the forces driving the transformation or removing the obstacles standing in their way [17].

Table 1 illustrates how emerging technology might affect its users and can be used to demonstrate a theory of change in the context of emerging technologies. There are five

fundamental processes in developing a theory of change. It should primarily describe all design elements a new technology possesses, the final effect it hopes to have on its users, and all potential outcomes that might lead to or assist in achieving this ultimate goal.

**Table 1.** Levels in a Theory of Change for emerging technologies in Education.

| No. | Levels |
| --- | --- |
| 1 | What effects may we anticipate from the new technology? |
| 2 | What are the emerging technology's intermediate results? |
| 3 | What are the anticipated emerging technology implementation activities? |
| 4 | What further tools or involvement are required to bring about change? |
| 5 | How will this change be planned for by the users? |

A theory of change table is constructive in impact evaluations because it makes it possible to pinpoint the steps that must be followed to achieve the desired educational results and the presumptions made along the way. Additionally, it ensures that the impact of developing technology is both measurable and may result from these results. The transparency that a theory of change brings into the problems that an emerging technology aims to tackle in education is the first step to take for impact evaluations.

This multidisciplinary theory of social change, culture, and human development draws ideas from anthropology, sociology, and psychology. It likewise has multiple layers and assumes that there are causal links between them. It includes sociodemographic factors, which have their origins in the German sociologist Tönnies (1957), at the top of the causal chain, cultural values at the next level down, and more conventional factors from developmental science at the following two levels—learning environment and individual development [18].

Figure 1 illustrates a paradigm of societal change, culture, and human growth. The terms "Community" and "Society" sum up the characteristics that serve as the foundation for each end of the sociodemographic dimension (top level of Figure 1). A small-scale social group with lifelong and close-knit relationships is referred to as a "community".

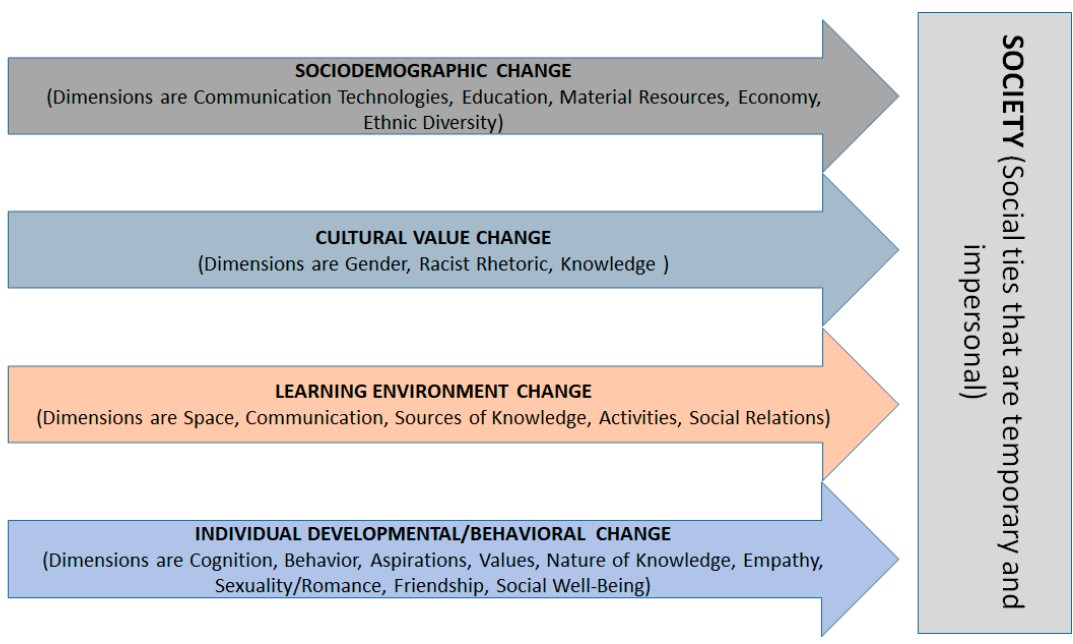

**Figure 1.** Model of social change, culture, and human development.

## 3. Research Methodology

Emerging Technologies for education are such a hot topic that many papers and conference proceedings are published in journals. The education sector is incorporating

emerging technologies into the classrooms; the aim is to enable more innovative and engaging teaching methods and learning experiences. It allows a flatter language lecture format, assists teachers in creating tailored learning content and provides access to ready-made content, and makes teaching activities more accessible.

### 3.1. Database Selection

Designed to obtain a comprehensive analysis of the qualified literature, we searched articles in major and well-renowned publications databases: WoS (Web of Science), Scopus, ScienceDirect, IEEE Xplore, MDPI, Google Scholar, and internet resources. The research was conducted on 565 research articles and publications, which were then processed a couple of rounds of filtering to remove the same articles collected in different databases and to synthesize the content of the selected articles. Thus, our search narrowed to focus on titles including "emerging technologies for teaching and learning", as illustrated in Figure 2 [19,20].

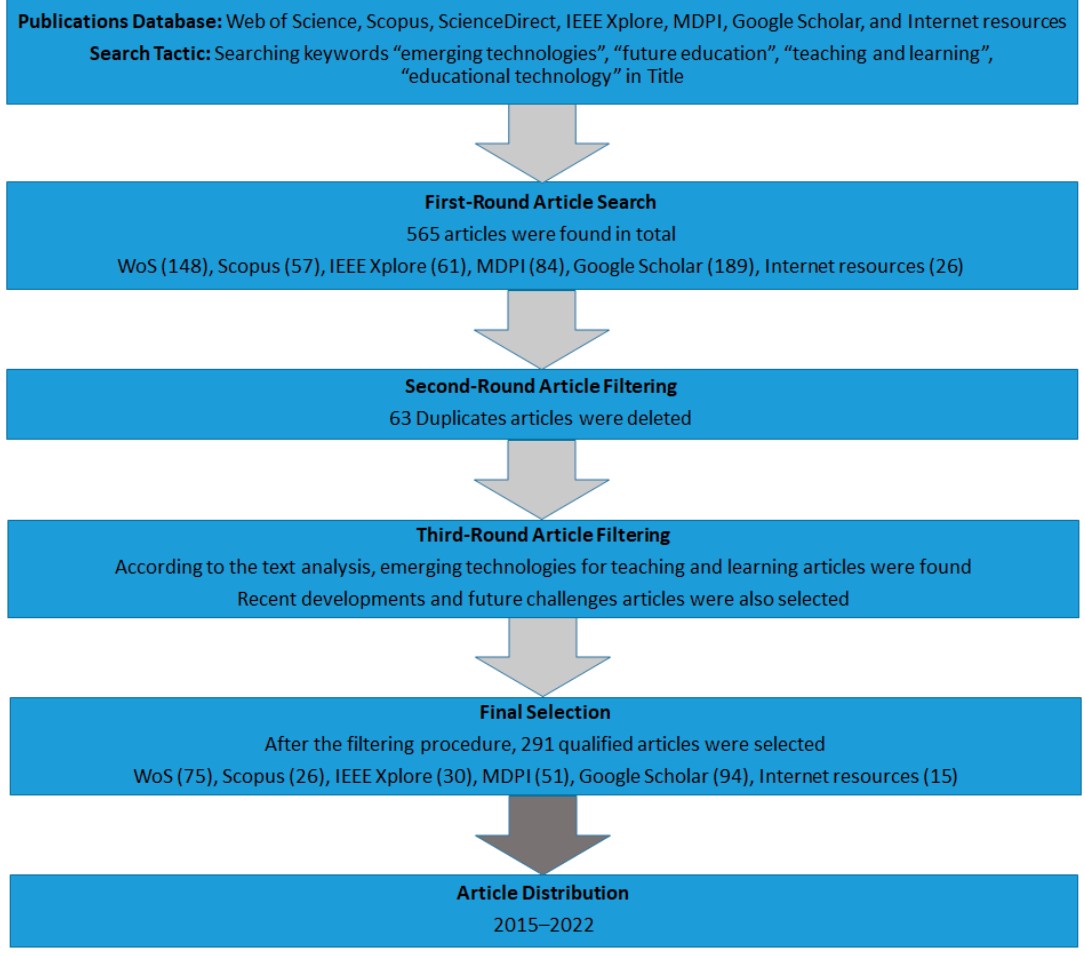

**Figure 2.** Research Taxonomy.

Scopus is one of the two big commercial bibliographic databases that cover scholarly literature from almost any discipline. In addition to searching for research articles, Scopus also provides academic journal rankings, author profiles, and an h-index calculator. Web of Science, also known as Web of Knowledge, is the second big bibliographic database. Usually, academic institutions provide either access to Web of Science or Scopus on their campus network for free. IEEE Xplore is the leading academic database in the field of engineering and computer science. It not only includes journal articles but also conference papers, standards, and books that can be searched for. ScienceDirect is the gateway to the

millions of academic articles published by Elsevier. In total, 2500 journals and more than 40,000 e-books can be searched via a single interface [21].

The researchers used a mix of keywords and addressed them in the following sections to conduct a thorough review of the literature for the current investigation.

### 3.2. Selection of Keywords

This multi-phase study critically analyzes peer-reviewed papers on new educational technologies. Multiple-phase searches and selections were made to find papers that could undergo comprehensive analyses. Even with clearly stated criteria, it is impossible to do a complete search given the explosion of online publications and open-access resources. This study was explicitly created to concentrate on research articles from several popular web-based databases. Additional search efforts were made to find more recent publications on future or innovative education specialty journals, as newer journal publications might not be included in databases. The source database was searched repeatedly using various keyword combinations, search techniques, and search terms such as "emerging technologies", "future education", "teaching and learning", and "educational technology". Additionally, the same search techniques were applied to research and development websites for publications published up to 2022.

The author and bibliographical keywords used in creating articles have been explored using VOSviewer. Additionally, using VOSviewer, researchers could create maps for the research items. Keywords for a nation or author are relevant topics for the investigation. Any two things have the potential to link strongly. Every link has a strength, which is shown by a positive numerical number; the higher this value, the stronger the linkage [22].

### 3.3. Articles Filtering

A set of selection criteria were devised and applied for inclusion and exclusion to accomplish the stated research goals. Only peer-reviewed, English-language journal publications presenting empirical, fact-based studies were chosen for further examination from the well-renowned publications databases. As a result of effectively putting an author's work under the scrutiny of other subject matter experts, peer-reviewed journal articles have grown to be the cornerstone of the scholarly publication system. The language that can most effectively cut across national boundaries and increase the impact of research is English since it is the lingua franca of science. To study abroad or read scientific publications as soon as they are published, one must be fluent in English. This allows one to use the publications only when they are necessary for their research projects and not just when they are available in their native language [23]. Thus, it encourages authors to strive to produce high-quality research that will advance the field. Additionally, the screening and selection process strictly adheres to the following criteria [24]:

1. Emerging technologies in teaching and learning environments must be the subject of research. Thus, published research on cutting-edge technology in non-educational settings such as engineering, the consumer market, healthcare systems, and others was disregarded.
2. Research must consist of empirical research with data. Articles that were primarily based on narratives or personal opinions were disqualified.
3. Research must examine how emerging technologies affect education by presenting pertinent qualitative data. Papers that offered no proof of learning were disqualified.
4. Additional papers for complete analyses included theoretical, conceptual, and literature reviews. These papers were carefully read to improve our background knowledge and to broaden the theoretical groundwork for constructing a general understanding of emerging technologies in teaching and learning.

### 3.4. Methods of Data Collection and Analysis

The primary literature, mostly scientific publications indexed in the top bibliographic databases, forms the basis of the primary study. As a result, the research methodology

used in this study is based on bibliometric methodologies, which enable a thorough examination of publications on "Promising Emerging Technologies for Teaching and Learning" at various levels. The proposed methodology is based on a quantitative analysis of all relevant publications that were chosen using keyword searches in the paper titles. The researchers looked over each article that qualified and studied it to determine the following: bibliometric, study setting, study nation, and trends in educational technology. The researchers included bibliometric information on the publication (such as the year of publication, the name of the journal, etc.), countries where studies were conducted, the study's educational context, and other information (e.g., K-12 or higher education). To develop shared understanding, the researchers talked, categorized, and discussed any themes that occurred during the process. The review was highly reliable and trustworthy because of the collaborative method used across many studies [24–27].

## 4. Emerging Technologies for Teaching and Learning

The effects of globalization and technological advancement on every aspect of human life are giving rising markets new opportunities and demands. To assist the growth of digital media literacy, it is becoming more and more crucial to include developing technologies in teaching and learning. The advent and subsequent usage of well-known technologies, such as Virtual Reality, Augmented Reality, Artificial Intelligence, the Internet of Things, and Cloud Computing in teaching and learning systems, have led to more dependable and effective learning environments as illustrated if the Figure 3. Globally, higher education is increasingly utilizing innovative technology. More and more lecturers are embracing new technology for casual and formal teaching and learning, often at the student's insistence. These technologies are ones that students already use in their social lives. Emerging technologies enable a personalized, adaptable, and differentiated focus on learning needs and pedagogy [28] and give learners more choices than a teacher-controlled, "one-size-fits-all" approach. Emerging technologies are tools, innovations, and breakthroughs used in various educational contexts to fulfill various educational purposes. Thus, institutional resources for funding, evaluating, and rewarding innovative pedagogical methods are prerequisites for using emerging technology to improve teaching and learning. For more than a decade, educational technologies have been developing at an exponential rate; however, many educational institutions struggle to comprehend and make use of innovations. To ensure that students receive a high-quality education, the authors introduce key emerging technologies in education that educators and institute administration must be aware of and comprehend.

### 4.1. Virtual Reality

Virtual reality, or VR, is a technology that generates a simulated environment through an artificial digital environment and an interactive computer-generated experience. With the help of this technology, a setting that resembles the real world can be created, or it can be a fantastical world that allows for experiences that are not feasible in the realm of traditional physical reality. The idea of virtual reality as we know it today was developed in the 1960s [29]. VR technology and education provide a novel form of learning that typically enhances the conventional approach. This innovative method helps to increase students' interest in learning in new ways that include sensory information. A more accurate display of functioning or activities can be achieved with virtual reality [30]. VR content will help students recognize and study abstract or difficult-to-observe knowledge in a risk-free environment, which is a crucial component of VR in education [31,32]. Students can access environments through computer simulations they would not otherwise have access to. Through interactive, three-dimensional computer-generated settings, virtual reality, or VR, aims to improve the realism of simulations significantly. Adding realism dramatically enhances the impact and general potency of instructional simulations. Additionally, VR might give engineering education access to previously unimagined capabilities.

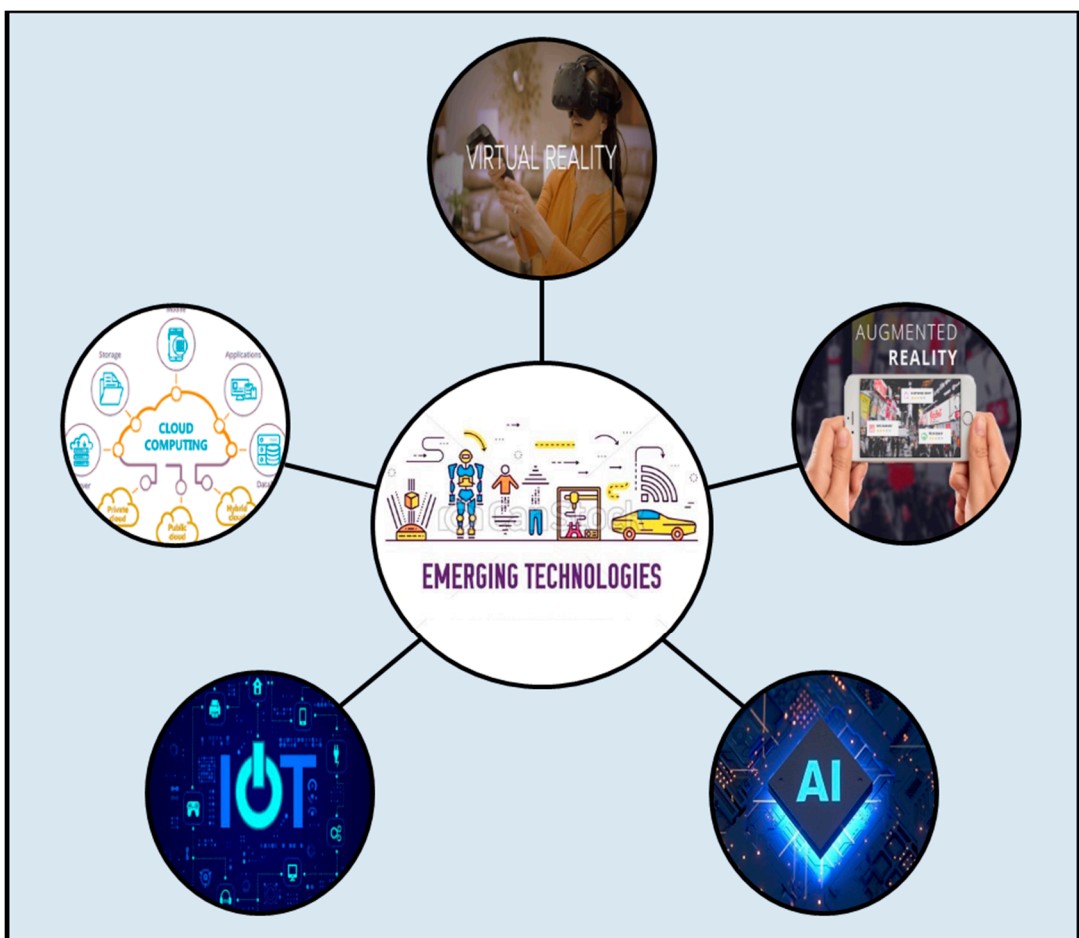

**Figure 3.** Prominent Emerging Technologies for Teaching and Learning.

4.1.1. Recent Developments

Virtual Reality's most immediately recognizable component is the head-mounted display (HMD). Human beings are visual creatures, and display technology is often the single biggest difference between immersive Virtual Reality systems and traditional user interfaces. Google Expeditions, a Google Cardboard platform, is a wireless fold-out smartphone-based cardboard viewer. The Expeditions are made up of various exciting tasks that can be used both inside and outside the classroom as extra homework or as a means of reviewing the topic [33,34]. The authors [35] describe a haptic interface-based immersive system that simulates task-specific training in a dangerous workplace. They employed an HMD backed by a movement-tracking device and feedback across numerous sensory (including touch) channels providing modules to enhance the realism of the simulation.

A VR program is presented [36] to help kids understand science and social topics. They employ Tilt Brush, which offers a 3D environment for painting, in their application. To simulate more realistic textures and input compared to conventional artificial teeth, haptic feedback was added to the Simodont [37] tools, a VR program for teaching crown fabrication in preclinical dentistry training. In addition to being used by a tutor who actively participates in the educational process, VR can also be used for self-study. In this instance, a natural person is teaching the class, and virtual reality is a technology that makes the class more engaging. Google Expedition serves as a good illustration of such a strategy. The authors [38] looked into the possible use of VR assistance for geography lessons. According to an examination of the lesson observations, the teacher stated that students ask more questions during lessons than during regular class times. In [39], they developed a VR platform to teach pre-university students about civil engineering through a VR game. The outcomes demonstrate that VR is valuable in civil engineering education since it enables

users to engage with the platform correctly, even without any prior experience. The authors describe a VR system that provides an interactive environment with a real-time 3D simulation of the heart's structure. The application enables specific interactivity, such as unrestricted manipulation, and models are disassembled to show the authentic anatomical relationships between the various heart components [40]. Virtual reality training is 3–4 times faster than conventional training, taking only 30 min to finish a session, according to the website for virtual medical coaching. Additionally, students who use virtual reality for skill development are more self-assured than those who employ conventional teaching techniques. Comparing VR training to traditional classroom training and e-learning, the effectiveness increases by 40% and 35%, respectively. Both students who received traditional medical education and those who received training in virtual reality (VR) were divided into two groups. According to the findings of a meta-analysis, VR-training pupils seem to pass exams more frequently than students who receive their education through more conventional means. The result showed a significant difference in the pass rate of the VR group and the traditional education group (Odds Ratio (OR) = 1.85, 95% confidence interval (CI): 1.32–2.58) [41].

### 4.1.2. Open Challenges

While new technologies have many benefits, it is unwise to consider them the solution to all our problems. Instead, we should address the issues they bring up head-on. It has been repeatedly demonstrated that VR is promising to improve educational outcomes by presenting a more stimulating environment that stimulates different perception points, such as visual perception, audio perception, olfactory perception, haptic (touch) perception, and gustatory (taste) perception [42]. Even though virtual reality has been studied for more than 50 years, there are still some reservations about its application. This section presents VR advancements in education and summarizes the most important issues and drawbacks of using VR technology as an educational tool. Modern VR solutions are based on head-mounted displays (HMDs), which offer absolute immersion through a 3D virtual environment that closely resembles reality. The lack of visual realism and the realism of the dynamics and interaction is one of the significant difficulties that need to be solved soon in the new features [43]. The current methods to create VR visuals and display technologies are relatively constrained. Be aware that the psycho–visual structure of the human brain enables us to recognize even subtle false elements, which can quickly shatter the immersion. Maximizing the appearance of reality in the VR world is thus a constant task. Several goods have warning labels for consumers because extended use may result in adverse health effects, such as motion sickness, disorientation, and loss of balance [44]. These result from high latencies, slow picture display rates, and high pixel persistence brought on by head movements while using an HMD. The first is connected to the risk that immersion can drastically lessen interpersonal communication, particularly between students. Interacting with a natural person is significantly more fulfilling than with an avatar. The same is true for screen interactions; they can never truly replace actual encounters. At all costs, student–teacher engagement must be preserved. Content delivery via technology and conventional classroom instruction needs to coexist harmoniously. The cost of these resources is the other restriction. To enable the simultaneous use of these bandwidth-hungry gadgets, institutes need a sizable IT and WIFI infrastructure. Lastly, training and education institutions find it expensive to invest in these new technologies (although costs are falling quickly) [45].

### 4.2. Augmented Reality

Augmented reality (AR) is one of the most prominent technology trends. It will only become more significant as AR-ready smartphones and other devices become more available worldwide. AR drew our interest because it creates a composite perspective by superimposing a digital image on a real-world setting. According to Charles McLellan, there are some significant distinctions between VR and AR, including "Although virtual reality (VR) offers the possibility of an infinite number of virtual worlds, the user's experience is

spatially constrained. When using a cutting-edge immersive VR system, you will be close to a powerful PC [46].

In contrast, AR nearly by definition necessitates that the user can move about in the natural environment with some degree of freedom" [46]. By providing a 3D picture of items, AR can improve conventional learning and improve comprehension. Consider medical students having the opportunity to open up a 3D model of a heart to see what the actual thing might look like [47]. AR is about improving comprehension by depicting objects that would be challenging to recreate in real life. It is not only about making the learning experience more fascinating and enjoyable. Several studies on augmented reality (AR) in education have come to the same conclusion: AR applications can improve the learning process, motivation, and effectiveness [48].

### 4.2.1. Recent Developments

Augmented reality (AR) has been defined as "systems that have the following three characteristics: (1) Combines real and virtual, (2) Interactive in real time, (3) Registered in 3-D" [47]. The simultaneous interaction between the actual and virtual worlds is made possible by augmented reality technologies. Digital content (text, audio, images, video, and 3D objects) is superimposed on the real world to give the impression that it is a part of it [49]. Internet of Things, smart watches, Google's Glass project, Microsoft's HoloLens, Facebook's Oculus Rift, bracelets, rings, necklaces, smart clothing, and tattoos are examples of wearable technologies that are frequently used in education. Early childhood education, primary and secondary education, elementary school through university levels, as well as diverse learner types at a kindergarten level, K-12 students, university students, adult learners, older adults, technical and vocational higher education, and students with special needs, have all been studied concerning the educational use of AR, with a focus on a variety of studies in the context of situated learning theory. These all came to the same conclusion: AR can provide a variety of learning possibilities with numerous advantages for both teaching and learning [50,51].

Wearable technologies include smartwatches, wristbands, sensor accessories, such as rings and necklaces, virtual reality glasses, the Google Glass project, its derived smart glasses, smart optical lenses, and headphones. The study shows the impact of using augmented reality books (AR books) on students' academic progress, and their perceptions of the surroundings were examined. The 22 participants in the trial received instruction from the researcher using the AR-based teaching materials created by the HITLibHZ-BuildAR application in a lab setting [52].

The use of augmented reality in medical education multimedia applications is a novel approach that has been researched. Because of this, it is challenging for students to understand topics such as neuroanatomy, which deals with the structure of the brain and blood arteries, in medical courses. This study focused on the use of 'mobile augmented reality applications in anatomy learning based on the benefits of AR technology and mobile learning approach in the field of education. The purpose of this study is to determine the medical faculty students' views on anatomy learning via 'mobile augmented reality technology [53]. A mobile application called Augment uses ARCore to display 3D models in augmented reality that are simultaneously scaled and placed in their actual surroundings. The authors looked into how this application affected the teaching of technical drawing [54]. The Aurasma application is one of the web 2.0 applications that makes use of augmented reality technologies. The Aurasma web 2.0 tool enables the free creation of interactive virtual reality content. These resources allow for more effective instruction and the delivery of beneficial information outside of the classroom. Aurasma was the first augmented reality-based application, and after 10 years of its lifetime, the popular augmented reality (AR) application was shut down and is no longer available.

### 4.2.2. Open Challenges

It is essential to look into potential fixes for reported issues with the technological aspects of AR applications (sensitivity trigger to recognition, GPS error, file size, etc.). Even though augmented reality technology has advanced, it can still be challenging for students to utilize; as a result, further research on the creation and use of augmented reality applications is required. In this manner, it is necessary to look into how students feel about usability and preferences in AR-based learning environments. Researchers should create holistic models and (empirically validated) design principles for AR settings to address the pedagogical challenges of AR [55]. Future research may look into the usage of AR applications to assist informal learning, collaborative learning, and ubiquitous learning, as well as the best ways to use them. In brief, it is possible to look into the effects of AR applications in more detail. To identify possible benefits, it is vital to research the integration of AR applications with new technology, such as vision glasses and educational results. By developing AR with varied demographics in mind, such as students with special needs, young children, and lifelong learners, its potential may increase further. It would be beneficial to look more closely at students' solutions to the issues they encountered when interacting with the environment in location-based augmented reality (AR) applications. To further understand the benefits of AR in educational contexts, more research might focus on student happiness, motivation, interactions, and engagement [56]. In the literature, some benefits and problems seem at odds with one another. For instance, some research said AR was challenging to use, while others claimed simplicity was beneficial. The same goes for whether AR applications lead to cognitive overload. Therefore, it is crucial to investigate the factors (topic, age group, interface characteristics, etc.) that contribute to the issue of cognitive overload in AR technology applications. More research is needed to understand how multisensory experiences relate to augmented reality applications and their effects on learning outcomes. The work of individuals who may desire to apply this technology in their future study would be facilitated by a thorough description of the materials development process and the considerations to be considered in design [57].

### 4.3. Artificial Intelligence

Researchers have improved computers' capacity for independent learning for more than 50 years, starting with the development of computers that needed application manipulation in the 1950s. This advancement marks a turning point in computer science, business, and society. In a way, computers have advanced to the point where they can finish new jobs independently [58]. Artificial Intelligence (AI) will communicate with applications using their native language, gestures, and emotions to adapt to them and learn from them. People will live in actual physical space and remain in the digital virtualized network due to the popularity and connectivity of various intelligent terminals. Since its inception, science fiction has long predicted the astounding and disastrous transformations that are supposed to result from the widespread use of artificial intelligence. Although sci-fi projected that AI would have a significant impact, it has quietly ingrained itself into many facets of our daily lives. In every industry, including education, AI is a major force for expansion and innovation in our daily lives. The impact of AI in education is being felt, and the conventional approaches are radically shifting. Emerging technologies are also changing how people teach and learn in education. Personalized learning, dynamic evaluations, and the ability to support meaningful interactions in online, mobile, or hybrid learning experiences are all attractive potentials of AI technology, which is flourishing in the education sector. Scholars have suggested replacing some teacher duties with robots or AI more provocatively, for instance, in response to the teacher shortage [59]. Research findings strongly suggest that embodied pedagogical agents (PAs) are required to facilitate effective education, despite the growth of online and screen-based PAs. Robots can play beneficial roles in education by addressing absenteeism, acting as catalysts for fruitful conversation during language instruction, offering emotional support to learners, and encouraging creativity and problem-solving, among other things. Robots in education have

also been considered conversational agents that engage in dialogue as well as intelligent tutoring systems (ITSs) or traditional robots [60]. Few studies discuss fully functional robotic teachers, suggesting a potential route for future development and research in AIED.

According to the eLearning Industry, up to 47% of learning management products will have AI capabilities activated. Everyone now has access to educational information via computers and smart devices, which has altered how people learn. AI is expanding quickly and has the potential to alter the educational landscape drastically. AI will assist in automating repetitive processes, such as fee collection, timetable creation, admission, etc., making school administration simple and open. Students will be able to learn anywhere, at any time, with the help of teacher-approved study material.

### 4.3.1. Recent Developments

Learning by teaching is a pedagogical strategy that has been digitally implemented by educational software that uses intelligent tutors or teachable agents (TAs). Intelligent tutors or teachable agents provide students with personalized, timely materials, guidance, and feedback. Research suggests conflicting results regarding its effects on learning despite its enormous potential. More recently, Swedish researchers looked at preschoolers' gaze patterns to determine how well they understood a math game based on TAs. Researchers concluded that TA was promising in supporting metacognitive scaffolding because the study revealed that young children regarded the TA as an independent entity [61]. Despite the extensive range of AI and machine learning applications, only a few research studies matched this study's requirements for comprehensive analysis. This powerful AI technology successfully identified changes in English as the second/foreign language (ESL/EFL) learning methods across various grades [62]. In a different study, information management practices of undergraduate students were utilized to predict their views toward educational uses of cloud-based mobile computing services with 74% accuracy [63]. The field of computer-assisted learning (CAL) develops alternatives using AI and digital technologies to support students' learning processes. AI can assist in mapping out each student's unique learning plans and trajectories, including their strengths and weaknesses, subjects that are more expensive but are simpler to understand or learn, and their learning preferences and activities. With the aid of teachers and schools, AI can personalize learning and enhance opportunities for students by using algorithms to guide them through various content paths [64]. Virtually limitless opportunities are opened up by AI technology for education. Researchers examined the effects of chatbot partners, as opposed to application partners, on students' course interest in foreign language classes with 122 students throughout a twelve-week experiment. After one week of using a chatbot, the study discovered that students' interests began to decrease. The Structural Equation Modeling results showed that task interest predicted future course interest for application partners but not for chatbot partners [65]. Researchers [66] recently looked into how elementary pupils in Taiwan were learning math and the impact of a picky expert system. Students in the experimental group in this study demonstrated greater learning achievement in mathematics than did pupils in the other two groups. The fifth-grade students in Taiwan who were anxious about arithmetic were also found to benefit from the adaptive learning model with affective and cognitive performance analysis. In an intelligent tutoring system [67], relating math to students' non-academic interests will boost learning, according to a study with high school students in the USA. As a result, highly personalized customization may help students succeed.

### 4.3.2. Open Challenges

The application of AI to teaching and learning has enormous promise. They inspire innovative research ideas and methodologies, take advantage of cutting-edge tools and technology for data gathering and analysis, and eventually become mainstream research paradigms. However, many academics and teachers still find them novel and unusual. Along with the new potential, we have also emphasized the significant problems and

development patterns in AI in education, mirrored in academic study, governmental decision-making, and business [68,69]. Some of the challenges of AI in teaching and learning are:

- Precision education and personalized learning are gradually replacing the one-size-fits-all model of education.
- The current concentration of AI research in education is on a particular field of intelligent computing technology.
- Machine-generated data should be carefully developed in terms of structure, intent, and meaning.
- Traditional formal education institutions are going through significant changes, possibly even a paradigm shift, in digitally driven knowledge economies.
- Creation of frameworks or models for AI-based learning.
- Analyzing how well students performed and how they found using current AI technologies.
- We are examining from many angles the efficacy of AI-based learning systems.
- Redefining and reexamining current conceptions of education in light of various uses of AI in the classroom.
- Advising on cutting-edge AI-supported learning or evaluating techniques.
- They are reevaluating and rethinking how to use the existing learning tools in learning content aided by AI.
- They are creating massive learning platforms, ethical guidelines, and best practices for using AI in educational settings.
- Cooperation between applications and AI.

An interdisciplinary study involving educators and educational researchers would likely produce workable, practical recommendations and excellent models for other educators. Additionally, collaborative research concentrating on AI technology applications that could directly or indirectly impact learning outcomes in actual educational settings is fundamental to realizing the full potential of AI in teaching and learning [24,70].

### 4.4. Internet of Things (IoT)

The Internet of Things (IoT) is a network of connected computing devices, mechanical and digital machinery, items, animals, or people that may exchange data across a network without requiring applications-to-applications or applications-to-computer interaction. IoT and the potential for rising profits, falling operating expenses, and growing efficiencies drive businesses. There has been increased interest in the IoT field from various industries. Consumer goods, business, health, manufacturing, education, research, and many more industries use IoT to enhance essential business processes. The education industry is also investigating cutting-edge digital infrastructure to enhance teaching and learning capabilities. Even for complex disciplines such as science, mathematics, engineering, etc., the Internet of Things has made it simpler for them to improve the teaching ecology. IoT devices can improve teaching methods in lecture halls and labs to more rapidly and effectively boost student interest [71].

This system makes education more interactive, communal, and widely accessible. Additionally, it makes interactive learning possible, ensures the safety of the educational facilities, boosts productivity, offers real-time learning experiences, allows for close supervision, etc. To build a sophisticated environment within universities, campuses must integrate IoT technology [72]. In reality, some of the most advantageous uses of IoT in education are the improvement of energy efficiency, exceptional communication between students and lecturers, and reduced operational costs. IoT will become more significant and capable of providing more insights when it is integrated into traditional teaching techniques. We may conclude that IoT is just converting traditional schooling into a digital paradigm [73].

### 4.4.1. Recent Developments

Technology impacts how people learn, live, play, and work. In terms of education, it has been crucial. IoT refers to many technologies and apparatuses that cooperate, and IoT changes how education is provided. Teachers may grade students more quickly by using online IoT methods, such as Kahoot and sharing grades with students via Telegram and Google Docs. Teachers provide students with helpful internet resources, such as information. They can assess the skill and knowledge of teachers as well, and management will look into the situation, keep an eye on things, and take the necessary action in light of the feedback given. Additionally, there are issues with managing students and staff, security, and excessive energy, heating, and water usage in universities without IoT devices [74]. The research team at The Hague Security Delta (HSD) Campus opened an Internet of Things forensic laboratory (IoT Forensic Lab) equipped with specialized hardware, software, and knowledge to enhance instruction. A good example of specialized gear that may be used to extract data from memory devices even when they are password-protected or have suffered fire or water damage is a chip-off. During their internship, teachers accompany bachelor students working with cutting-edge digital forensic tools in the IoT Forensic Lab [75]. To incorporate the fundamentals of IoT and blockchain technologies in STEM education, this research [76] created a learning kit. The learning kit gives students experience by utilizing a project-based learning strategy. The kit has components for the "brain," "muscle," and "cloud" to address every aspect of IoT and blockchain technologies. To educate students about the Fourth Industrial Revolution, it is crucial to introduce IoT and blockchain concepts to them. This article [77] suggests a framework for creating an Internet-of-Things (IoT)-enabled "smart campus" for universities. The Internet-based connection between controls, sensors, and physical things is embraced by the IoT infusion in education. Inspiring new interactions between objects and people, the IoT paradigm sets a wide range of conditions for institutions. It also identifies the potential to make them smarter. The goal of this research is to create smart parking spaces, smart classrooms, and smart learning environments for students.

A blended learning (BL) paradigm with an IoT foundation may be the greatest New Normal option for all stakeholders in the education sector during this COVID-19 pandemic. Therefore, to stop COVID-19, traditional face-to-face (F2F) is being forced to adapt because of social distance. In this study [78], BL was classified into three types of learning environments—Digital, Embedded, and Side-by-Side cases—each of which was further subdivided into four characteristics: F2F, Selfpaced, Tele-D, and Ubiquitous. Using relevant literature, textbooks, research, articles, and websites, a model was analyzed and synthesized using the content analysis method. This model's framework links six modules, a collection of databases, two types of contexts, and two roles of user interfaces (teacher and student) (classroom and personal). The issues surrounding green IoT methods in engineering education are used to construct smart classrooms [79]. The sustainability of IoT resources is emphasized in engineering education, which encourages everyone involved in the institution to act sustainably. According to the author, several chores must be completed to do this. These tasks are (i) the best possible use of resources; (ii) disposing of, reusing, and recycling IoT materials; and (iii) raising awareness of green IoT among educational institution stakeholders. By completing these objectives, future IoT resources would be more sustainably produced.

### 4.4.2. Open Challenges

IoT can be an effective educational instrument for developing training skills in students who use technology to solve unsolved challenges. Additionally, wearable technology enables students to monitor and record their academic behavior, which enhances their interactive learning environment. In contrast, there are several difficulties related to using IoT in education [71]. To qualify their recent graduates with the necessary skills to manage and work on various IoT projects, departments must adapt their curricula to include IoT courses. Including IoT in developed curricula, planning and organizing orientation

sessions for staff on the benefits of IoT, providing professional development plans for teachers, and raising awareness among students of the various IoT applications are all examples of improved strategies for educational institutions. In addition, IoT is still in its early phases, and other problems, including wireless coverage, expensive sensors, and battery life, remain unresolved [80]. To make it easier for IoT applications to be used, IoT engineers and developers must consider these concerns. Although several IoT-based studies have explored m-learning applications such as augmented reality and learning analytics, their adoption still needs improvement and more research. Many academics believe that security and privacy issues are among the biggest obstacles to adopting IoT in education. Thus, to lessen the difficulties found in earlier studies and maintain its usefulness, future efforts to deploy IoT in education must consider these variables.

Furthermore, although the use of Green ICT in education has been extensively researched in many affluent nations, its adoption in developing nations is still underutilized. To assist the decision-makers in developing effective methods for its deployment, scholars need to look into the elements influencing its adoption. Last but not least, it is asserted that wearable technology, such as Google Glass, is widely employed in medical education. Researchers must therefore investigate how these technologies are applied and adopted in different fields [81].

### 4.5. Cloud Computing

"Cloud Computing" refers to a system where all data processing and storage takes place outside the user's devices and through these databases. A distributed computing application that offers scalable and dynamic computer resources, such as storage and processing power, is supplied as a service over the Internet. Nowadays, practically every area of information technology uses cloud computing due to its simplicity in deployment, management, scalability, security, and other related qualities. Instead of installing applications on their premises, which burdens them with the costs of building networks, maintaining, and managing them, it is practical and advantageous for enterprises to use cloud data centers [82].

Most private educational institutions now rely heavily on information technology to meet their needs. More often, professors and students can access these services through web browsers and Internet technology. The services are provided for free or at a far lower education cost, frequently with considerably higher availability than the educational institution can supply. Through the Internet and cloud computing, users may manage and access data. The institution's primary users are all connected to the cloud, and each user has a separate login for their assigned tasks. Students can access all of the teaching materials offered by the teachers through the Internet using computers and other electronic devices at home and school, 24 h a day. Teachers can submit class tutorials, assignments, and assessments on the cloud server. By examining students' study logs, the educational system will enable teachers to pinpoint problematic areas where students tend to make mistakes. This will enable educators to develop their instructional strategies and materials. This will enable educators to enhance their resources and pedagogical strategies [83].

#### 4.5.1. Recent Developments

Traditional online learning methods, such as live instruction and multimedia, need to meet the needs of contemporary experimental education. Virtual reality (VR) is a simulated environment where computer graphics create a realistic-looking virtual world that may react to a user's input. A cloud-to-end rendering and storage system was presented [84] to address these problems and offer high-quality educational experiences with minimal latency. They separated the experimental scenes into two categories: interactive models and background. Due to its capacity to enhance students' learning processes, gamification has drawn much interest in innovative teaching techniques. The authors [85] conducted a study and provided solutions. The principal objective was to lay out the overall plan for creating a gamification environment. Seven requirements—motivation, objective clarity, testing

concepts, cheating scenario monitoring, task optimization for long-term improvement, and overall lose/win results of games—were shown to be useful for instructional purposes. Using cloud computing technologies, this research [86] has tried to address technical problems, including learning module upgrades and scalability. It can use the infrastructure to develop games with educational and motivating objectives.

Cloud computing has dramatically facilitated online collaborative learning engagements in education. They discovered that collaborative learning tasks involving editing, sharing, discussion, and communication were carried out using cloud computing tools. A poll of 170 IT students at a private Malaysian institution revealed they were willing to keep using cloud e-learning applications for their coursework [87]. The students underwent testing for one trimester to evaluate their exposure to cloud-based e-learning tools. M-learning is a component of the educational process that makes learning more accessible and offers a practical way to improve education's value significantly. The authors [88] built a multimodal, interactive, cloud-based mobile learning platform that enables learners to participate according to their preferred learning styles. Utilizing functional adaptations to fit the learning objectives of children with specific needs, this tool increased their learning capacity by about 30%. This study [84] creates a cloud-to-end rendering and storage system with two models: background and interactive, for experimental VR education. Background rendering is done on the cloud server, and the finished product is sent as a video stream to the end terminal. After that, at the end terminal, interactive models are light-rendered and blended. To increase image quality when the user's point of view changes, a newer 3D warping and hole-filling method is also suggested.

4.5.2. Open Challenges

Although there are obvious benefits to using cloud computing in education, several hazards have also been highlighted in the research that has been examined. Before adopting and using cloud computing in educational contexts, these hazards should be considered as they differ for educational stakeholders. Sensitive data protection is crucial in the educational sector, and cloud computing's handling of this issue is particularly relevant. Some comments have raised the possibility that cloud computing can protect these data more securely than conventional distributed systems. They contend that data are kept on virtual servers that are inaccessible to criminals, that harmed services can be replaced more quickly and cheaply, and that security and monitoring are centralized and can be handled more successfully [89]. Service-level agreements are strictly adhered to for all business applications. Operational teams are crucial to administering service-level agreements and application runtime governance. The operational teams in production environments support appropriate clustering and failover data replication system monitoring (Transactions monitoring, logs monitoring, and others), Maintenance (Runtime governance), Recovery from Disaster performance monitoring, and capacity. A cloud provider could suffer substantial harm and negative impact if they cannot adequately supply any of the services mentioned earlier. In any case, to guard against security threats such as packet sniffing or traffic analysis, educational institutions should examine how data are safeguarded during transmission and storage. To boost user confidence, security audits and certifications should be carried out. Institutions should consider employing specialist security services such as encryption or single sign-on capabilities even though cloud providers provide safe infrastructures [90]. Although there are many cloud service providers, platform and infrastructure management are still in their infancy. For many businesses, features such as "Auto-scaling" are an essential requirement. The current scalability and load-balancing features have much room for improvement. To prevent the single point of failure caused by security assaults, it is advised to hire more than one cloud provider to host educational services and data, especially given that these attacks are more frequently directed toward pertinent public cloud providers [91]. Government rules in various European nations prohibit the physical storage of sensitive information and personal data about customers outside of the state or nation. Cloud service providers must set up a data center or storage facility only within the

nation to meet these requirements and abide by laws. For cloud providers, having such an infrastructure may only sometimes be possible and is a significant problem. Ad hoc agreements might be made to avoid these issues if some cloud providers currently need to guarantee legal compliance [92].

## 5. Conclusions and Future Works

The literature on cutting-edge techniques integrating ICT makes it abundantly evident that emerging technology strategies aimed toward learning are being pursued to produce a beneficial environment for teaching and learning. In total, 81% of the students in higher education institutions believe that digital learning tools, such as virtual classroom chat, improve their marks, according to a Statista study [93]. The education industry's fastest-growing segment is e-learning, which has increased by 900% since 2000 [94]. According to a recent Smoothwall research, 96% of instructors think that using developing technology in the classroom will increase students' engagement and learning. In total, 56% of respondents think that pupils are noticeably more engaged when technology is used in the classroom [95].

According to the study's findings, emerging technologies do undoubtedly improve pedagogical practice, notably in terms of quick feedback, teamwork, and engagement between teachers and students. Educational institutions would need to consciously create governance structures and strategic plans for integrating the use of these technologies into institutional life to fully realize the potential that emerging technologies have for enhancing teaching and learning practices. Google Expeditions is a wireless fold-out cardboard viewer that can be used inside as well as outside the classroom, serving as an additional review of the material or homework [30,31]. The Simodont [27] is a Virtual Reality application for teaching crown fabrication to simulate more realistic textures and feedback as compared to traditional false teeth. The Augmented Reality-based HITLibHZ-BuildAR program has been used frequently in medical education [49]. Augment applications to visualize 3D models in Augmented Reality applications affected the teaching of technical drawing, integrated in real-time in their actual size and environment [51]. Some academics have publicly suggested using artificial intelligence robots to replace instructors or certain tasks in the teaching profession [60]. In addition to the intelligent tutors and teachable agents in online or blended learning, as reported in AIEd studies [57]. Teachers save time by using Kahoot online IoT processes to share marks with students via Telegram and Google Docs [71]. A blended learning paradigm with an IoT foundation may be the greatest new normal option for all stakeholders in the education sector during this COVID-19 pandemic [75]. In [81], they proposed a cloud-to-end rendering and storage system to mitigate live instruction issues and provide high-quality educational experiences with low latency. A private Malaysian university also supported the student's willingness to continue using cloud e-learning applications in their studies [84]. The expansion of emerging technologies for educational research would lead to new approaches to teaching and learning as well as more useful guidelines and examples for educators.

The study emphasizes the growing necessity to address the obstacles preventing the seamless integration of emerging technologies into teaching and learning. The outcome confirms that integrating emerging technologies in the twenty-first century is crucial to achieving teaching and learning goals. The study's findings make it clear that emerging technologies for teaching and learning can help to update obsolete materials and techniques employed in conventional teaching and learning processes, which frequently elevate the teacher to the role of expert or the only source of knowledge. For educators and students to fully benefit from the potential advantages of emerging educational technologies, all stakeholders in the education sector must take prompt action to address the identified impediments that prevent the successful integration of emerging technologies into the teaching and learning process. More research is needed to determine the skills required today and forecast potential skill shortages and gaps.

**Author Contributions:** Conceptualization, A.A. and M.A.; methodology, A.A. and M.A.; software, M.A.; validation, A.A. and M.A.; formal analysis, M.A.; investigation, A.A. and M.A.; resources, A.A. and M.A.; data curation, A.A. and M.A.; writing—original draft preparation, M.A.; writing—review and editing, A.A. and M.A.; visualization, A.A.; supervision, A.A. and M.A. All authors have read and agreed to the published version of the manuscript.

**Funding:** This research received no external funding.

**Institutional Review Board Statement:** Not applicable.

**Informed Consent Statement:** Not applicable.

**Data Availability Statement:** The data are available from the corresponding author on reasonable request.

**Conflicts of Interest:** The authors declare no conflict of interest.

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
