# Peer review of "Promising Emerging Technologies for Teaching and Learning: Recent Developments and Future Challenges"

_sustainability, doi:10.3390/su15086917_

Round 1

Reviewer 1 Report (Previous Reviewer 1)

- clear abstract

- some more background information is provided with more sources.

- better theoretical and research design presentation but still sources are missing for a couple of points.

- there are still some issues with the use of the language (connection of sentences); I suggest proofreading should be used

- informative discussion enriched with studies, data and evidence

- clear conclusion

- interesting and up-to-date sources

Author Response

- clear abstract

The abstract has been revised.

- some more background information is provided with more sources.

An additional literature review has been added from lines 98 to 120.

- better theoretical and research design presentation but still sources are missing for a couple of points.

Missing sources have been added.

All comments on the pdf file provided by the reviewer have been resolved.

Confusing; rephrasing is needed (rephrased, lines from 44 to 47)

qualified in terms of what? (explanation of qualified teacher has been added, lines from 54 to 56)

how is big population a barrier to education and how is it related to teacher's qualifications? (sentence has been rephrased with proper justification, lines 53 to 54)

Sources based on which these criteria have been selected? (source has been added, line 253)

Sources based on which this data analysis method has been used? (sources/references has been added, line 282)

Not proper register for an academic paper (rephrased, line 371)

Unclear (explain the details of intelligent tutors and teachable agents, line 523 to 526)

- there are still some issues with the use of the language (connection of sentences); I suggest proofreading should be used

Professional proofreading has been done.

- informative discussion enriched with studies, data and evidence

The introduction section has been revised with the latest literature review.

- clear conclusion

The conclusion and future work has been revised according to the reviewer's comments.

- interesting and up-to-date sources

Missing sources have been added to the manuscript.

Reviewer 2 Report (New Reviewer)

Dear authors, with pleasure I got acquainted with your article. It deals with important topical issues of our time.

However, a number of remarks

1. I recommend paying closer attention to the annotation

It is the first thing that is read in the article, while it does not become clear from the annotation:

  Goals and results

• It is not clear from the annotation what results you come to.

• You write that "The analysis focuses on peer-reviewed research papers in well-renowned publications databases, drawing upon a biblioetric analysis." (Lines 20-21)

 • What is the novelty of your article?

• The goal is blurred, without specifics? Very general.

2. Methods (lines 240-257, and lines 20-22) are not relevant to the stated purpose

3. Also…

• Line 125. What are "Good theories"? You start Theoretical Concept with this concept. What are you putting into it? Why are they "Good"?

Good luck

Author Response

Dear authors, with pleasure I got acquainted with your article. It deals with important topical issues of our time.

 However, a number of remarks

  1. I recommend paying closer attention to the annotation

It is the first thing that is read in the article, while it does not become clear from the annotation:

  Goals and results

  • It is not clear from the annotation what results you come to.
  • You write that "The analysis focuses on peer-reviewed research papers in well-renowned publications databases, drawing upon a bibliometric analysis." (Lines 20-21)
  • What is the novelty of your article?
  • The goal is blurred, without specifics? Very general.

The abstract has been revised as per the reviewer’s comments.

  1. Methods (lines 240-257, and lines 20-22) are not relevant to the stated purpose

The methodology has been revised and Figure 2 of the research taxonomy added aligns with the research goal and results.

  1. Also…
  • Line 125. What are "Good theories"? You start Theoretical Concept with this concept. What are you putting into it? Why are they "Good"?

The definition of Good Theory has been added from lines 142 to 145.

Reviewer 3 Report (New Reviewer)

The authors cite in too much detail results (statistical data) of the researched articles without making analog with other similar studies. There are no formulated generalized results and conclusions. The purpose of such study is to summarize and systematize the results achieved in the area of research.

When presenting the research methodology, the authors can explain the process of selecting the articles more systematically and precisely - it is not clear how many articles meet the preliminary selection, how many remain after the subsequent selection and finally the number of studies that are the subject of analysis.

Other notes:

The figures are not suitable for this kind of research, because they do not correspond to discussed questions, connections, processes.

The authors use abbreviations without prior explanation (TA, ESL, EFL - p.12).

Applications that are no longer supported and out of date (eg Aurasma) are discusses - the authors may note that they are no longer supported and not available.

Author Response

The authors cite in too much detail results (statistical data) of the researched articles without making analog with other similar studies. There are no formulated generalized results and conclusions. The purpose of such study is to summarize and systematize the results achieved in the area of research.

The conclusion and future work has been revised according to the reviewer's comments.

When presenting the research methodology, the authors can explain the process of selecting the articles more systematically and precisely - it is not clear how many articles meet the preliminary selection, how many remain after the subsequent selection and finally the number of studies that are the subject of analysis.

Figure 2 Research Taxonomy has been added to the manuscript.

Other notes:

The figures are not suitable for this kind of research, because they do not correspond to discussed questions, connections, processes.

Most figures have been removed as per the reviewer’s suggestion.

The authors use abbreviations without prior explanation (TA, ESL, EFL - p.12).

The abbreviation has been added.

Applications that are no longer supported and out of date (eg Aurasma) are discusses - the authors may note that they are no longer supported and not available.

Comment has been added, lines 454 to 456.

Round 2

Reviewer 2 Report (New Reviewer)

The article has been significantly improved in contrast to its first version.

The authors listened to the recommendations and significantly revised the abstract, the introduction methodology, expanded the part regarding the review of sources, and also added modern sources to the review. In addition, a part of Conclusions and Future Works, i.e., conclusions, has been significantly worked out.

I believe that in this version the article can be recommended for publication

Reviewer 3 Report (New Reviewer)

Thanks to the authors for considering the critical remarks.

This manuscript is a resubmission of an earlier submission. The following is a list of the peer review reports and author responses from that submission.

Round 1

Reviewer 1 Report

- Clear abstract

- Interesting and up-to-date sources

- Interesting and nice analysis but certain points must be addressed (check comments in the paper)

- Language issues (grammar, weak sentence connections, lack of linking words, not academic style/register in some cases)

- Sources are missing for some points

- The theory is not analysed adequately

- Methodology: it needs more information and justifications as to why certain choices have been made

- Need to search about more studies (check relevant comment in the paper)

- Clearer conclusions are offered

Reviewer 2 Report

Why is teaching and learning given a short form of L & D?

The introduction is weak, vague and it lacks substantial update literature support. Claims such as “On the other hand, tools and technologies can meet the needs of all educational levels, regardless of place and time” reflect more of a propaganda rather than critical reading of literature. Claim like “Although there have been several studies on teaching and learning, none specifically focus on the use of new technologies in teaching and learning” is  completely unfounded.

The research questions are too broad to be answered properly. Any one form of technologies would require very substantial review work for the past 5 years (assuming this is the newness).

Figure 2 is not theory at all. They are questions and should end with question marks.

The search term ““emerging technologies for teaching and learning" cannot be accepted as high quality searching strategies from the perspective of library science. This is arguably the most important step for a good literature review, which the manuscript fails to achieve.

The findings are well known (for e.g., the five emerging technologies). The analysis was not well presented or insightful. For e.g., “The most significant concerns and disadvantages of using VR technology as a teaching tool while presenting VR advancements in education. Most current VR solutions are based on head-mounted displays (HMDs), which offer absolute immersion through a 3D virtual environment that closely resembles reality. The lack of visual realism and the realism of the dynamics and interaction, according to [26], is one of the significant difficulties that needs to be solved soon in the new features”.

The overall quality does not warrant revision. 

Reviewer 3 Report

This is a review article concerning promising emerging technologies for teaching and learning, which aims to examine the promising emerging technologies, their most recent advancements, and the obstacles they will face in the future. Three questions were addressed: 1) What kind of research is being published in specialized journals about new technologies for teaching and learning? 2) According to admissible research articles, what are the applications of emerging technologies and their educational advantages? 3) What are the most recent advancements in emerging technologies for education, and what obstacles lie ahead?

Recommendation: Major revision

This topic selected by this review is potentially meaningful given the increasingly vital role of various technologies, especially the emerging ones, in education. The authors have in particular given a detailed illustration of the five types of emerging technologies. However, the manuscript presents some severe problems that weaken its impact and value. My comments are listed below.

The first problem concerns the Abstract, which in my opinion should be rewritten. An abstract is a short summary of the whole research, and its main task is to concisely report the aims, methods, and outcomes of your research, so that the readers could know exactly what the paper is about. It is thus clear that your current version of abstract is poorly structured. Almost two thirds of this section was used to introduce the background information of the study, which however should be the job of the Introduction. This leads to the fact that the aims, methods, and findings of the study was not reported adequately.

The second problem arises from the research questions. Although there were three research questions listed in the Introduction (Line 79-84), it appears that the first two were not tackled at all (or at least not clearly addressed) in the subsequent sections. This constitutes a major flaw of the design of this review.

The third problem arising from the Research Methodology section lies in the current lack of a set of systematic review procedures. In other words, many details that should be specified in the article so as to make it a qualified systematic review were omitted. For example, the exact search combinations used by the authors should be given, the specific inclusion and exclusion criteria for article filtering should be described in more detail, and a coding scheme or framework should be included so as to clearly illustrate how exactly the five types of emerging technologies were identified and how their recent developments and future challenges for learning and developing were found.

Except for the above major problems, a number of problematic expressions are found in the paper. For instance:

a) Humans are visual creatures, the most significant distinction between immersive Virtual Reality systems and conventional user interfaces is frequently the display technology. (Line 222-223).

b) The most significant concerns and disadvantages of using VR technology as a teaching tool while presenting VR advancements in education. (Line 262-263)

Besides, the abbreviation of a term should be given when the term is first appeared in the main body of the paper. Appropriate citations should be provided whenever a quotation is referenced (e.g., Line 303-305).

The authors are recommended to carefully read the manuscript to avoid these problems.